# Enhancing Essential Grains Yield for Sustainable Food Security and Bio-Safe Agriculture through Latest Innovative Approaches

Ghosoon Albahri [1], Amal A. Alyamani [2], Adnan Badran [3], Akram Hijazi [1], Mohamad Nasser [1], Marc Maresca [4,*] and Elias Baydoun [5,*]

1 Doctoral School of Sciences and Techonolgy-Platform for Research and Analysis in Environmental Sciences (EDST-PRASE), Beirut 1107, Lebanon; ghosoonalbahri@gmail.com (G.A.); akram.hijazi@ul.edu.lb (A.H.); mohamednasser@hotmail.fr (M.N.)

2 Department of Biotechnology, College of Science, Taif University, P.O. Box 11099, Taif 21944, Saudi Arabia; a.yamani@tu.edu.sa

3 Department of Nutrition, University of Petra, Amman 11196, Jordan; abadran@uop.edu.jo

4 Aix Marseille Univ, CNRS, Centrale Marseille, iSm2, 13013 Marseille, France

5 Department of Biology, American University of Beirut, Beirut 1107, Lebanon

* Correspondence: m.maresca@univ-amu.fr (M.M.); eliasbay@aub.edu.lb (E.B.)

**Abstract:** A key concern in agriculture is how to feed the expanding population and safeguard the environment from the ill effects of climate change. To feed a growing global population, food production and security are significant problems, as food output may need to double by 2050. Thus, more innovative and effective approaches for increasing agricultural productivity (hence, food production) are required to meet the rising demand for food. The world's most widely cultivated grains include corn, wheat, and rice, which serve as the foundation for basic foods. This review focuses on some of the key most up-to-date approaches that boost wheat, rice, corn, barley, and oat yields with insight into how molecular technology and genetics may raise the production and resource-efficient use of these important grains. Although red light management and genetic manipulation show maximal grain yield enhancement, other covered strategies including bacterial-nutrient management, solar brightening, facing abiotic stress through innovative agricultural systems, fertilizer management, harmful gas emissions reduction, photosynthesis enhancement, stress tolerance, disease resistance, and varietal improvement also enhance grain production and increase plant resistance to harmful environmental circumstances. This study also discusses the potential challenges of the addressed approaches and possible future perspectives.

**Keywords:** essential grains yield; effective approaches; food security; sustainable agriculture

## 1. Introduction

One of the numerous challenges of modern agriculture is the need for a significant increase in output to meet the needs of an expanding human population [1]. The principal grains that are cultivated worldwide are rice (*Oryza sativa*), maize (*Zea mays*), wheat (*Triticum aestivum*), and barley (*Hordeum vulgare*) [2]. Before the COVID-19 pandemic, there was a spike in food insecurity; in 2019, it was projected that 25.9% of the world's population (or around 2 billion people) had moderate to severe food insecurity, up from 22.4% in 2014. Additionally, grain output has been affected by harsh weather conditions for several years [3].

Food availability fundamentally relies on food production and distribution, both of which are essential elements in ensuring food security. Food demand is rising, and the food supply is being gradually constrained by factors such as growing urbanization, land degradation (such as salinization and erosion), non-food use of crops and cropland (such as bioenergy and recreational activities), and climate change. Grain crop yields and the supply of food must be enhanced to ensure global food security. This calls for integrated, diverse,

and sustainable strategies to improve grain resource use efficiency while also increasing productivity per unit area [4].

The history of wheat domestication and utilization is strongly related to human attempts to regulate our food supply and battle hunger. Wheat is now cultivated on a global scale and is one of the primary food sources [5]. In temperate regions, wheat is the most significant staple crop, and demand for it is rising in nations that are urbanizing and industrializing [6]. In addition to wheat, rice (*Oryza sativa* L.) is farmed in a variety of ecosystems, including those that are subjected to floods and droughts, and more than half of the people on Earth eat rice as their primary dietary source [7]. Moreover, maize (*Zea mays* L.) was initially domesticated about 9000 years ago from a wild relative called the lowland grass teosinte [8]. *Zea mays* is a main food source, where the genetic changes and natural variations have contributed to the evolution, diversification, and conservation of the species [9]. Maize is a cereal grain that is grown extensively around the world and ranks third in terms of cereal grain importance after rice and wheat in terms of global consumption. Maize kernels can be white, yellow, orange, red, or black depending on environmental, cultural, and genetic factors [10]. One of the earliest cereals to be domesticated was barley (*Hordeum vulgare* L.), which has been used as the main food source in regions where other grains are more difficult to grow [11]. It has been demonstrated that the general metabolite composition of barley and its subsequent malt can be influenced by both genetics and the environment [12,13]. Oats are currently grown on a global scale and are a key component in the diets of many people [14].

Since ancient times, it has been known that higher grain crop yields are necessary to feed growing global populations. Even though the world's population has expanded by around eight times since 1800 to about 8 billion people in 2019, plant breeders have been able to enhance yields to keep up with this growth [15]. Estimates suggest that 800 million people still experience calorie deficit despite the relatively high yields of modern plant types, and this is a rising issue given that the world's population is anticipated to reach around 11 billion people by the end of this century. As a result, agricultural yields must continue to rise, but plant breeders' rates of yield growth are slowing down and are no longer sufficient to keep up with population expansion [16]. Additionally, anthropogenic climate change means that plants will have to endure higher levels of abiotic stresses to thrive [17]. Several technologies have played a significant role in raising plant yields over the past century, and others will continue to play an important role in future plant improvements [18]. There is a need to review the most up-to-date innovative agricultural approaches that enhance grain productivity, varieties, and disease resistance. This review focuses on variety of advanced agricultural practices, including (i) wheat bio-stimulant bacterial formulations, nutrient inoculations, light management, abiotic stress adaptation, and rust resistance; (ii) rice gas emissions reduction, system complexation, fertilizers combinations, and molecular manipulation; (iii) corn plant photosynthesis enhancement, solar brightening, and molecular breeding; (iv) barley leaf architecture changes and disease resistance; and (v) oat innovative models and fungi resistance.

This review summarizes the latest innovative yield-increasing approaches to improve wheat, rice, corn, barley, and oat productivity in order to ensure food security, minimize hunger, and empower agricultural strategies (Figure 1), demonstrates the potential challenges accompanying present agricultural technologies, and illustrates promising future perspectives.

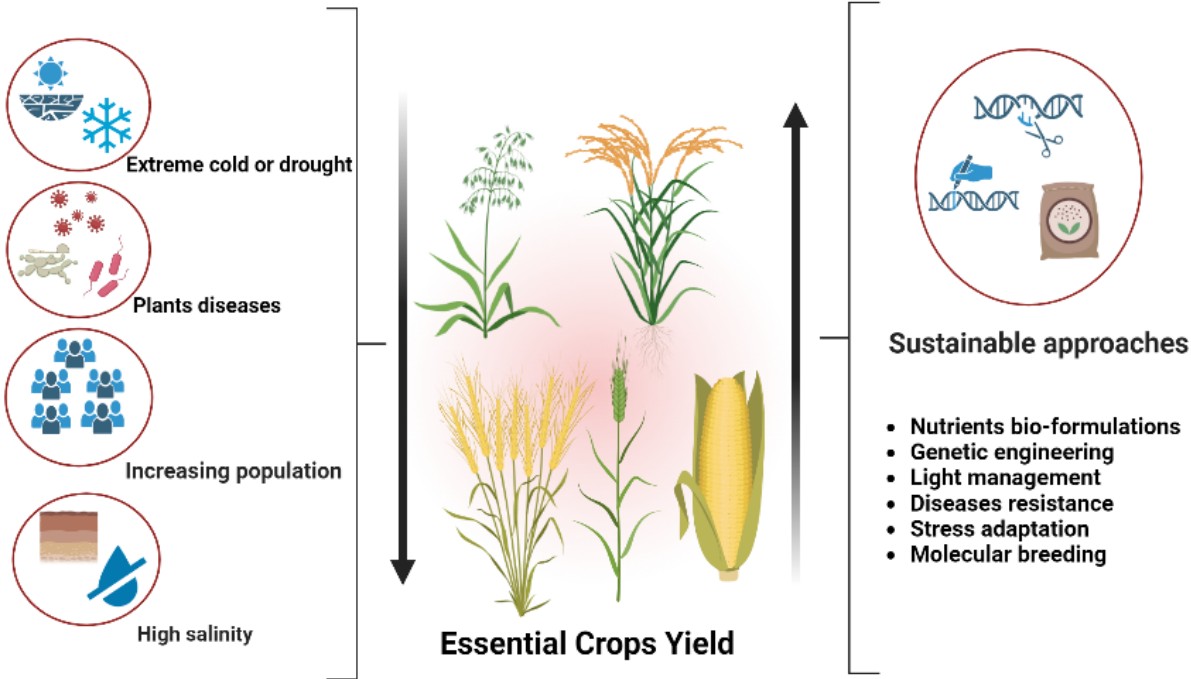

**Figure 1.** Major obstacles that decrease and solutions that increase yields of essential crops (wheat, corn, rice, oats, and barley).

## 2. Methodology of the Review

Principal research database sources were utilized to search the literature for relevant publications, including Scopus (Elsevier), PubMed (National Institutes of Health (NIH), Springer Nature (SN), Frontiers, and MDPI. Keywords were identified that covered the main ideas investigated, such as "essential grains", "wheat", "rice", "corn", "barley", "oats", "yield increase", "innovative approaches", "bacterial formulations," "fertilizers", "agricultural practices", "light management", "genetic engineering", "molecular breeding", "disease resistance", "photosynthesis enhancement", "varieties improvement", "challenges", and "perspectives". The main steps followed for the study were: (1) create the main objective and goal of this study, (2) perform the suggested key searches in the global research databases, (3) collect data from recent articles, (4) prioritize articles published in high impact factor (primarily Q1 and Q2) journals, and (5) create figures to visually illustrate the approaches discovered during the review. Since ancient times there have been plenty of agricultural practices ranging from the oldest techniques to the most recent innovative environmental and molecular strategies, but this study focused on the more common among them.

## 3. Wheat

About 8000 years ago, the tetraploid *Triticum turgidum* and the diploid *Aegilops tauschii* were hybridized to produce wheat (*Triticum aestivum*, genome AABBDD) [19]. The genome of wheat underwent quick modifications throughout this evolutionary process, resulting in currently farmed wheat lacking several genes compared to its ancestors [20]. The most widely cultivated grain on the planet, wheat (*Triticum aestivum* L.), contributes to nearly a fifth of the total calories consumed by people and has the highest protein content of any meal [21]. Breeders work to create better varieties by genetically adjusting complicated yield and quality factors while preserving yield stability and geographic tolerance to biotic and abiotic challenges [22]. Wheat yield needs to increase by more than 60% while preserving or even improving its nutritional qualities in order to feed the 9 billion people who are expected to be alive in 2050. Emphasis must be placed on critical features related to plant productivity and response to environmental difficulties to accomplish this

aim without needing to increase the area of cultivated land, which is simply not possible. Meanwhile, a deficiency in this important staple grain could pose a severe danger to global food security [23]. The most efficient weed-control method for wheat is chemicals; however, the fast evolution of herbicide-resistant weeds threatens food security and necessitates the use of non-chemical methods [24]. The discovery of important dwarfing genes, which are now widely used in breeding programs around the world, made this easier. The potential yield performance of the top wheat cultivars is severely hampered by several impediments. However, the development of several new innovative approaches involving molecular genetics and biotechnology, including bacterial formulations, nutrient management, light control, smart agricultural practices, and biotechnological strategies, have facilitated the replacement of difficult, resource-intensive, and time-consuming cytogenetic investigations and enhanced wheat yield as shown in Figure 2 [25,26]. Quantitative resistance, which is controlled by minor genes, has a complicated genetic base and is active against all patho-types and races of disease-causing pathogens. Adult plant resistance (APR) is frequently identified as field resistance because adult plant resistance (APR) is most effective at the post-seedling and adult plant phases. The Sr2 (stem rust resistance gene) and Lr34 (leaf and stripe rust and powdery mildew resistance gene) are the two most well-known APR genes in wheat [27]. Ug99 has a very broad spectrum of virulence toward most R genes and quickly evolved virulence to crucial R genes (Sr24 and Sr36), which hampered the first emergency breeding program to include resistance to this strain. This complexity was further increased by Ug99 [28].

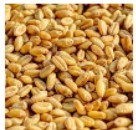
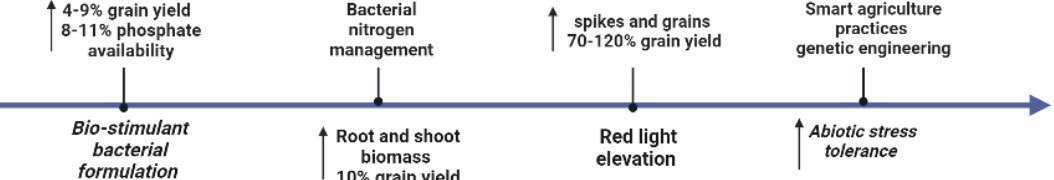

**Figure 2.** New approaches for the sustainable increase in wheat production.

### 3.1. Synergistic Bio-Stimulant of Wheat Production by Autochthonous Phosphate_Solubilizing Bacteria Formulation

To achieve sustainable agriculture, rhizobacteria (PGPR) stimulate plant growth and stress tolerance. This method is an environmentally friendly and sustainable alternative to synthetic agrochemicals and traditional agricultural methods [29]. Phosphorus has a big impact on wheat crop growth and output. Phosphorus is necessary for normal growth and good yield of wheat crops in the early stages. The growth of wheat roots and seedling establishment are improved by an adequate amount of phosphorus [30]. There is a huge potential for sustainable agriculture with the use of phosphate-solubilizing bacteria (PSB) as bio-fertilizers. Recent research explored rock PS by *Ochrobactrum* sp. SSR (DSM 109610) by linking it to bacterial gene expression and looking for an effective formulation. The expression of pyrroloquinoline quinone biosynthesis protein C (pqqC) and phosphate levels were six times higher in SSR-inoculated soil that had been treated with rock phosphate (RP) than in uninoculated control soil. Moreover, four organic additives, biochar, compost, filter mud (FM), and humic acid, were combined to create bio-formulations with the two well-characterized PS, *Enterobacter* spp. DSM 109592 and DSM 109593. The FM-based bioformulation was extremely effective and increased both seed P (9%) as well as wheat grain yield (4–9%). Additionally, soil phosphate availability (8.5–11%) and phosphatase activity (4–5%) were improved by the FM-based bioformulation [31].

### 3.2. Improving Sustainable Field-Grown Wheat Production by Azospirillumbrasilense Nitrogen Management

To maximize profitable production while minimizing adverse environmental effects, managing nitrogen (N) inputs in wheat production systems is a crucial challenge [32]. Wheat and other small grain crops are heavily reliant on nitrogen fertilization, which

leads to a higher concentration of crude protein and higher nutritional value in these crops than in C4 crops. With a 50-year average of 52 kg ha$^{-1}$ year$^{-1}$, wheat production uses 18% of the world's nitrogen fertilizer [33]. Research showed that inoculation with *Azospirillum brasilense* may be a key sustainable management practice to improve wheat production in tropical environments because of its low economic cost, ease of application, and the high likelihood that wheat crops will respond favorably and will associate with various N application levels. This procedure has the potential to raise the yield of wheat grains, the utilization and uptake of nitrogen, and overall farm profitability. Additionally, the inoculation of wheat plants with *A. brasilense* boosted agronomic efficiency, apparent nitrogen fertilizer recovery, and nitrogen uptake, as well as root, shoot, and grain nitrogen (N) accumulation at both ordinary and high N application levels. Additionally, inoculation enhanced root and shoot biomass, resulting in a 10.3% rise in yield when compared to plants that were not inoculated [34].

### 3.3. Maximizing Wheat Yield by Increasing Exposure to Red to Far-Red Light during Specific Stages of Wheat Development

Under continuous illumination with eight distinct light qualities (in eight plots) produced by mixing three of four different types of fluorescent lamps (white, blue, purplish red, and ultraviolet-A), the development rate of wheat was examined at a constant temperature of 20 °C. The findings implied that regardless of the photoperiod and vernalization, green and red lights were significant regulators of the developmental pace [35]. Based on recent findings, the yield sensitivity to a high ratio of red to far-red light (R:FR) varied between positive if applied after the initiation of stem elongation and negative or null if applied before, with two cultivars showing yield increases of up to 70% and 120%, respectively. These yield increases were correlated with more tiller spikes and grains per spike in the main shoot. Obtaining a high R:FR from stem elongation to maturity is a promising step toward a considerable increase in grain production, which indicates that R:FR exerts a major control on wheat yield potential [36].

### 3.4. Negotiating Smart Agricultural Practices to Increase Wheat Yield

Due to long-standing practices, the agricultural sector is simultaneously contributing to several significant environmental problems, including air pollution from high levels of greenhouse gases (GHGs). This continuous addition of GHGs to the environment is responsible for the drastic changes in climate that are known as climate change [37]. Climate change has a significant impact on the agricultural industry because crop yield and performance are mostly determined by environmental conditions [38]. Various farming techniques ranging from traditional to cutting-edge technology are being implemented all over the world to increase agricultural productivity. Each has advantages and disadvantages. The goal of one such strategy, smart agriculture practice (SAP), is to deploy smart agricultural techniques that will produce more yield with fewer GHG emissions [39]. To boost wheat crop output, smart agriculture techniques such as subsoiling have been applied, which is advised in the context of climate change in agriculture. The subsoiling methods used include subsoiling ploughing tillage (SPT), subsoiling harrow tillage (SHT), and no-tillage subsoiling (SNT). A vibrating subsoil trowel is used to plow the ground, subsoiling it to a depth of 14 to 14.5 inches. Subsoiling preserves soil nitrogen content by preserving soil moisture content (SMC) and soil organic carbon (SOC), which permits a limited release of NOx from the soil. It was discovered that higher pH promotes less COx absorption into the oil and less NOx emission from the soil, but higher SOC induces more COx absorption into the soil and more NOx emission [40].

### 3.5. Improving Wheat Adaptation to Abiotic Stresses through Biotechnical and Genetic Approaches

Finding novel regions in the wheat genome that can increase yield is essential for enhancing food production. An APETALA2/ethylene responsive factor (AP2/ERF) transcription factor called DUO1 has recently been shown to be amenable to gene editing.

Researchers used CRISPR-Cas9 to modify the gene and created wheat plants with many spikelets in the lower center section of the spikes. Furthermore, live imaging revealed that the basal spikelet primordia of the gene-edited wheat had more and larger cells than the wild type, which indicated that the gene was likely involved in controlling cell division. Field testing revealed that the gene-edited wheat plants produced more grains per spike than the wild variety, indicating an increase in production [41]. The activation of downstream genes that respond to various abiotic stressors is aided by transcription factors. As a result, genes that encode transcription factors that respond to abiotic stress are a useful target in genetic engineering research aiming to increase plant resistance to various harmful environmental circumstances [42]. The majority of transcription factors that have been effectively employed to increase wheat's tolerance to drought come from the DREB/CBF, ERF, NAC [43–49], HD-ZipI, and WRKY families, as well as the ABA-stress-ripening (ASR) transcription factor (TaASR1-D), which is involved in drought tolerance through ABA signaling, the BES/BZR transcription factor, and nuclear factor Y [50,51]. Additionally, recent research identified potential genes related to yield, including Rht-B, Rht-D, and TaMFT, in recombinant inbred lines by combining linkage mapping and weighted gene co-expression network analysis (WGCNA). Plant height (PH), spike length (SL), and seed characteristics underwent quantitative trait loci mapping (QTL). Additionally, QTL TaSL1 was found with several effects on kernel length and SL regulation, which can be used to improve wheat production. These findings offered useful molecular markers and gene data for future precise mapping and cloning of yield-related trait loci [52].

### 3.6. Utilizing Genetic Rust Resistance in Wheat and Integrated Rust Management Techniques to Create More Resilient Cultivars

Fungi and insects reduce global wheat production by 21.5% each year [53]. Rust diseases, which are among the most economically significant diseases impacting wheat production, are caused by biotrophic pathogenic fungi. The rust pathogens *Puccinia graminis* f. sp. *tritici*, *Puccinia triticina (pt),* and *Puccinia striiformis* f. sp. *tritici*, which cause stem rust, leaf rust, and stripe rust, respectively, put the world's wheat production at risk all year [54–56]. Wheat leaves at different developmental stages, as well as the leaf sheath and glumes, are the main targets of *pt* infection. Several genes, known as Lr, Yr, and Sr, confer resistance to leaf rust, stripe rust, and stem rust, respectively [57,58]. However, despite significant advancements in the treatment of wheat diseases through scientific and technological innovations, plant diseases continue to pose serious dangers to the world's wheat output [59,60]. A more thorough understanding of the spatial and temporal heterogeneity in the growth pattern of wheat rust will contribute to more efficient and long-lasting disease control [61]. However, the ability to anticipate the severity of wheat leaf rust based on the climate (relative humidity and temperature), the disease itself (onset and severity), its genotype (maturity and resistance), and planting date is still limited [62]. Wheat TaWAK6, a non-arginine-aspartate wall-associated kinase with an extracellular GUB domain, a calcium-binding epidermal growth factor domain, and a cytoplasmic serine/threonine kinase domain, was demonstrated to be significant for the establishment of quantitative and mature plant resistance [63]. It was demonstrated that TaRPM1, an NBS-LRR gene in wheat, favorably regulates the salicylic acid signaling system, which confers *Pst* resistance in high-temperature plants. During plant–pathogen interactions, silicon-induced biochemical or molecular resistance predominated, involving the activation of defense-related enzymes, regulation of the intricate network of signal pathways, stimulation of the production of antimicrobial compounds, and activation of defense-related genes [64,65]. Wheat stripe rust treatment techniques have advanced significantly, but more research is needed to understand how the pathosystem's environment, disease, and planting date affect resistance [66]. Because planting date has also been regarded as a key disease management strategy for a wide range of grains, the close association between wheat maturation and wheat leaf rust predictions seems significant. Furthermore, given that a combination of the environment, maturity, and planting date would be investigated for screening wheat varieties, these findings may improve breeding

for more resistant genotypes [67]. These connections need to be further investigated in other geographic regions with different host and pathogen genotypes, climatic conditions, and environmental situations.

## 4. Rice

Nearly half of the world's population relies primarily on rice as a food supply and source of energy, which has important consequences for both nutrition and health [68]. One of the most significant historical developments was the domestication of cultivated rice (*Oryza sativa* L.) [69]. Since rice is the second-largest food crop in the world, rapid climate change brought on by human activity has a significant impact on rice. For rice to better adapt to uncertain surroundings, research on the evolution of several rice ecotypes is crucial [70]. More than one-third of the world's population, 90% of whom live below the poverty line, depend on rice as a major food source [71]. Thus, the genetic advancement of rice is crucial for boosting socioeconomic benefits and minimizing the environmental effects of agriculture, in addition to being significant for ensuring the world's food security [72]. Scientists are using genes from several rice types to increase the resilience of rice to pests, diseases, and environmental stress as part of ongoing attempts to address the long-standing issues of food security and sustainable agriculture [73]. Molecular mechanisms underpinning distinct rice phenotypes and analysis of the underlying regulatory networks, encompassing omics, genome-wide association studies, phytohormone action, nutrient uptake, biotic and abiotic responses, photoperiodic flowering, and reproductive development (fertility and sterility), have advanced dramatically since completion of the rice reference genome [74]. According to recent findings, applying biochar enhanced rice yield and nitrogen use efficiency by 10.73% and 12.04%, respectively, and for enhancing rice production and NUE, application rates of >20 t/ha biochar and 150–250 kg/ha nitrogen fertilizer were suggested. The application of biochar during water-saving irrigation increases rice yield more effectively, according to a study on water management [75]. Several approaches have been incorporated successfully in the improvement of rice yield, including bio-fertilizer combinations, de-escalating methane emissions, and genetic strategies, as shown in Figure 3.

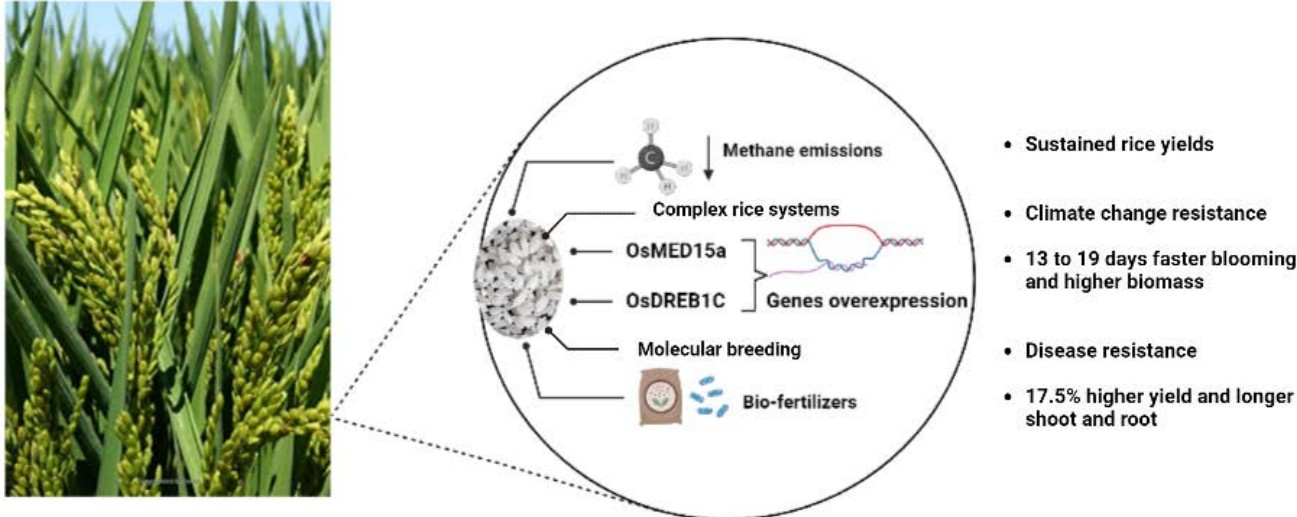

**Figure 3.** Recent innovative approaches that contribute to rice production improvement.

### 4.1. Varying Fertilizer Combinations and Planting Techniques Result in Different Rice Yields

In the tropics, fertilizer dominates the process of producing rice. Many rice farmers employ chemical fertilizers, which are thought to be a superior approach by quickly enhancing rice productivity [76]. The adverse effects of extensive chemical fertilization are now a major issue on a global scale. Low fertilizer efficiency is primarily the result of

ineffective fertilizer management. Due to the high temperatures in the tropics, intensive and high-dose nitrogen fertilizer application can cause nitrogen to leach and volatilize. In tropical soil, phosphorus adsorption is a significant barrier to adequate plant uptake. In addition, the Russia–Ukraine war is causing chemical fertilizer costs to rise quickly. Thus, the use of biofertilizer (BF) and organic matter amendment is advised to reduce the use of chemical fertilizers [77,78]. The effectiveness and productivity of rice production systems using broadcast seeding and transplanting under various fertilizer combinations were recently compared in an experiment. The control treatment consisted of NPK fertilizer and urea, while the subplots included mixtures of biofertilizer (BF) with and without compost. Both nitrogen-fixing bacteria and phosphate-solubilizing microorganisms were present in the BF. The fertilizer mixture enhanced the number of tillers, 1000-grain weight, shoot height, root length, shoot and root dry weight (RDW), and root-to-shoot ratio. When rice was transplanted and fertilized with urea and NPK fertilizer, the yield was 17.5% greater than when rice was broadcasted. A comparable yield was obtained by combining chemical fertilizer, compost, and BF; the yield of transplanted rice was higher than that of broadcasted rice by 2.18% [79].

*4.2. Maintaining Sustainable Rice Yield through Methane Emissions Reduction from Rice Cropping Systems*

Rice cultivation is criticized because it contributes significantly to methane ($CH_4$) emissions. A novel strategy for sustainable agriculture may involve the use of substances that control plant development and have the potential to boost crop yields while reducing $CH_4$ emissions [80,81]. The development of ethephon (2-chloroethylphosphonic acid), a precursor of ethylene ($C_2H_4$), as a potential soil amendment for lowering methane ($CH_4$) emissions was studied [82]. Ethephon was combined with biodegradable polymers such as cellulose acetate and applied to paddy soils to decrease the release of $C_2H_4$. According to the findings, using cellulose acetate-mixed ethephon reduced $CH_4$ emissions in rice paddies and maintained sustainable yields [83]. Other findings showed that regular use of sulfur-coated urea in combination with uncoated urea could preserve rice output while lowering methane emissions from rice paddies [84]. Moreover, it was proven that water stress could lower $CH_4$ and $N_2O$ emissions for some rice types, and planting drought-resistant rice varieties under water-shortage irrigation might be able to sustain rice yields and limit the potential for global warming [85].

*4.3. Increasing Rice Yield and Yield Stability Despite Changing Weather Conditions by Complex Rice Systems*

It is anticipated that climate change will increase weather variability and the frequency of extreme weather events such as floods and droughts [86,87]. In addition to possibly being hazardous to the environment and human health, increased usage of agrochemicals to address this issue also makes farmers more dependent on outside supplies. Although it depends on the type of organic system, organic agriculture has been suggested to lessen harmful effects on the environment and human health [88]. However, it is frequently linked to lower crop yields and labor-intensive crop management techniques [89–91]. Contrary to monoculture systems, complex rice systems (CRSs) can be more productive. In CRSs, the cultivation of Azolla, fish, and ducks is integrated into the rice production system, which has been demonstrated to be a potential method for addressing pest issues in organic rice production systems while also increasing rice yield [92]. It has been demonstrated that the integration of Azolla, fish, and ducks produced durable systems that retained high yields under heavy rainfall [93]. Additionally, research demonstrated that the rice cropping system's ability to adapt to harsh weather events was improved by the addition of additional features [94]. Thus, a possible model for climate change resistance in rice production and food security preservation in Asia and beyond is the complex rice system design [95].

### 4.4. Improving Rice Crop Yield through Integrated Molecular Genetics Approach

The size (length and width) of the grain affects its weight. Among these characteristics, grain size/weight has been the focus of many breeding initiatives aimed at increasing rice productivity [96]. The presence of two paralogs of MED15 in rice and a rise in OsMED15 1's expression level (referred to as OsMED15 a) at various stages of seed development suggested a potential role in the process. However, OsMED15b expression in transgenic plants had no impact on seed size, development, or yield, supporting the significance of OsMED15a in rice [97]. Rice grain size/weight-regulating TFs are linked to their target genes through the mediator subunit OsMED15a. OsMED15a may be implicated in rice seed development, according to expression analyses and high-resolution quantitative trait loci (QTL) mapping. Previous research found that many grain size/weight-related SNPs in OsMED15a, OsNAC024, and OsNAC025 in 509 low- and high-grain-weight rice genotypes. These genes all showed enhanced expression during seed development. The grain size/weight-regulating genes GW2, GW5, and DR11 were downregulated by RNAi-mediated repression of OsMED15a expression, which also decreased grain length, weight, and yield [98]. This approach will help to direct efforts to introduce useful alleles through marker-assisted introgression in rice crop improvement. In addition, recent findings detected a member of the DREB (dehydration-responsive element binding) family, OsDREB1C. OsDREB1C's expression is influenced by both light and low nitrogen status, and it activates a variety of transcriptional programs that regulate flowering time, nitrogen uptake, and photosynthetic ability. OsDREB1C is a transcriptional activator that is found in the nucleus and cytoplasm and directly binds to cis regions in DNA, such as the dehydration-responsive element (DRE)/C repeat (CRT), GCC, and G boxes. At the genome-wide level, 9735 putative OsDREB1C-binding sites were discovered using transcriptomic and chromatin immunoprecipitation sequencing (ChIP-seq) methods. It was discovered that five genes targeted by OsDREB1C [ribulose-l,5-bisphosphate carboxylase/oxygenase small subunit 3 (OsRBCS3), nitrate reductase 2 (OsNR2), nitrate transporter 2.4 (OsNRT2.4), nitrate transporter 1.1B (OsNRT1.1B), and flowering locus T-like 1 (OsFTL1)] are closely associated with photosynthesis, nitrogen utilization, and flowering, and these key traits are altered by OsDREB1C overexpression. Moreover, DNA affinity purification sequencing (DAP-seq) and ChIP-quantitative polymerase chain reaction (ChIP-qPCR) experiments demonstrated that OsDREB1C binds to the exons of OsNR2, OsNRT2.4, OsNRT1.1B, and OsFTL1 to stimulate the transcription of these genes. Field tests with rice overexpressing OsDREB1C produced a yield that was 41.3 to 68.3% greater than that of wild-type plants, with more grains per panicle, heavier grains, and better harvest indices. Additionally, OsDREB1C-overexpressed plants showed improved photosynthetic ability and increased nitrogen absorption, transport activity, and grain output, particularly in low-nitrogen environments. In addition, these plants bloomed 13 to 19 days earlier and accumulated more biomass at the heading stage than wild-type plants. Thus, they had better nitrogen usage efficiency and encouraged optimal resource allocation, offering a promising plan for achieving much-needed gains in rice agricultural production [99].

### 4.5. Enhancing Rice Crop Productivity by Sustainable Perennial Rice Approach

Annual crops mostly cover the soil intermittently with plants, leaving it vulnerable to severe rain that removes large amounts of soil and nutrients, thus diminishing its fertility. In conserving crucial ecosystem services, perennial crops have an advantage over annual ones. Since perennial crops typically have longer photosynthetic seasons, their production is increased, and more light is intercepted each year. The deep roots of perennial crops have been demonstrated to promote soil carbon accumulation and nitrogen retention in addition to providing persistent living cover, which may ultimately result in lower rates of fertilizers, pesticides, and labor input. Perennial cropping methods enhance farmers' livelihoods while also benefiting the natural systems necessary to sustain productivity over the long term by enhancing labor productivity and soil quality [100,101]. Recent studies showed that *Oryza sativa*, a domesticated annual rice plant native to Asia, was crossed with

*Oryza long staminate*, a perennial rice plant native to Africa. In contrast to 6.7 Mg of replanted annual rice, which needed more work and seeds, irrigated perennial rice yielded grains from a single planting for eight consecutive harvests over the span of four years, averaging 6.8 Mg ha$^{-1}$harvest$^{-1}$. With the aid of the perennial rice, soils accumulated 0.95 Mg ha$^{-1}$yr$^{-1}$ of organic carbon and 0.11 Mg ha$^{-1}$yr$^{-1}$ of nitrogen, and their pH increased by 0.3–0.4 and plant-available water capacity by 7.2 mm. Farmers strongly favor perennial cultivars because they reduce labor expenses by 58.1% and input expenditures by 49.2% throughout each regrowth cycle. Thus, perennial rice is a revolutionary crop that can be adapted to a wide range of frost-free climates between 40° N and 40° S. It has the potential to increase livelihoods, improve soil quality, and stimulate research on other perennial grains [102].

*4.6. Increasing Rice Blast Disease Resistance through a Molecular Breeding Approach*

The most significant constraints on rice production are diseases. There have been reports of over 70 diseases affecting rice that are brought on by fungi, bacteria, viruses, or nematodes [103]. The most severe rice disease is rice blast (*Magnaporthe oryzae*), which is particularly destructive when conducive conditions are present, and since there have been numerous outbreaks of blast disease in rice, attempts have been made to create new cultivars that are resistant to the disease [103,104]. To create an integrated management program for blast resistance, strategic research is currently focused on bridging the gaps in our understanding of the effects of biotic stresses on rice, particularly by advancing our understanding of the molecular genetics of blast disease. The identification of potential DNA markers associated with resistance genes using fine mapping may enable rice breeders to effectively transfer these genes from donor cultivars into new, elite rice cultivars via marker-assisted selection (MAS) [105]. There are many uses for allele mining in grain improvement, including allele identification, allelic variation characterization, haplotype identification, analysis of haplotype diversity between related or different haplotypes of the same gene, evolutionary relationships, and similarity analysis, as well as the creation of molecular markers to distinguish one allele from other alleles [106]. The goal of allele mining is to find the best alleles for blast resistance in rice species that are both wild and domesticated. Numerous important blast resistance genes have been found to have unique and superior alleles in both wild and farmed rice varieties using mining techniques. The resistance and susceptible alleles of Pi54 have been distinguished using functional markers generated using allele mining approaches, which have also examined the sequence level similarity of Pikm alleles. Allele mining has also been used to differentiate *M. oryza* from *M. grisea* [107,108].

## 5. Corn

After rice and wheat, maize (*Zea mays* L.) is the third-most significant food crop in the world. Tropical maize has long been a foundation of the diet of Sub-Saharan Africa; 95% of the maize farmed is consumed directly as human food and serves as a significant source of income for the rural population, which is poor in resources [109]. The most abundant cereal in the world is corn, which is utilized as fuel, food for livestock, and human consumption [110]. Along with rice and wheat, maize is one of the "big three" grain species that account for more than half of the calories consumed globally. In addition, maize has served as a model system for plant genetics and cytogenetics from the beginning of the study of genetics in the early 20th century [111]. Maize yields have increased almost sevenfold since the creation of single-hybrid breeding programs in the early half of the 20th century, and a large portion of that gain can be attributed to the tolerance of higher planting densities. There are 1800 genomic areas that present targets of selection in contemporary breeding and 160 loci-mediating adaptive agronomic characteristics used in genome-wide association and selection scan approaches [112]. A recent study offers conceptual support for the development of a quantitative design for a maize yield that is high yielding while being sustainable. The maximum leaf area per plant varied with plant density, according to the results. The best maximum leaf area per plant, according to yield model estimation,

was 0.63 times greater than the hybrids' potential maximum leaf area per plant, and the yield performance equation demonstrated outstanding prediction with a satisfactory mean RMSE value of 7.72% [113]. Moreover, nitrogen-doped carbon dots and solar brightening are two of the latest approaches that have contributed to the improvement of maize yield, as shown in Figure 4.

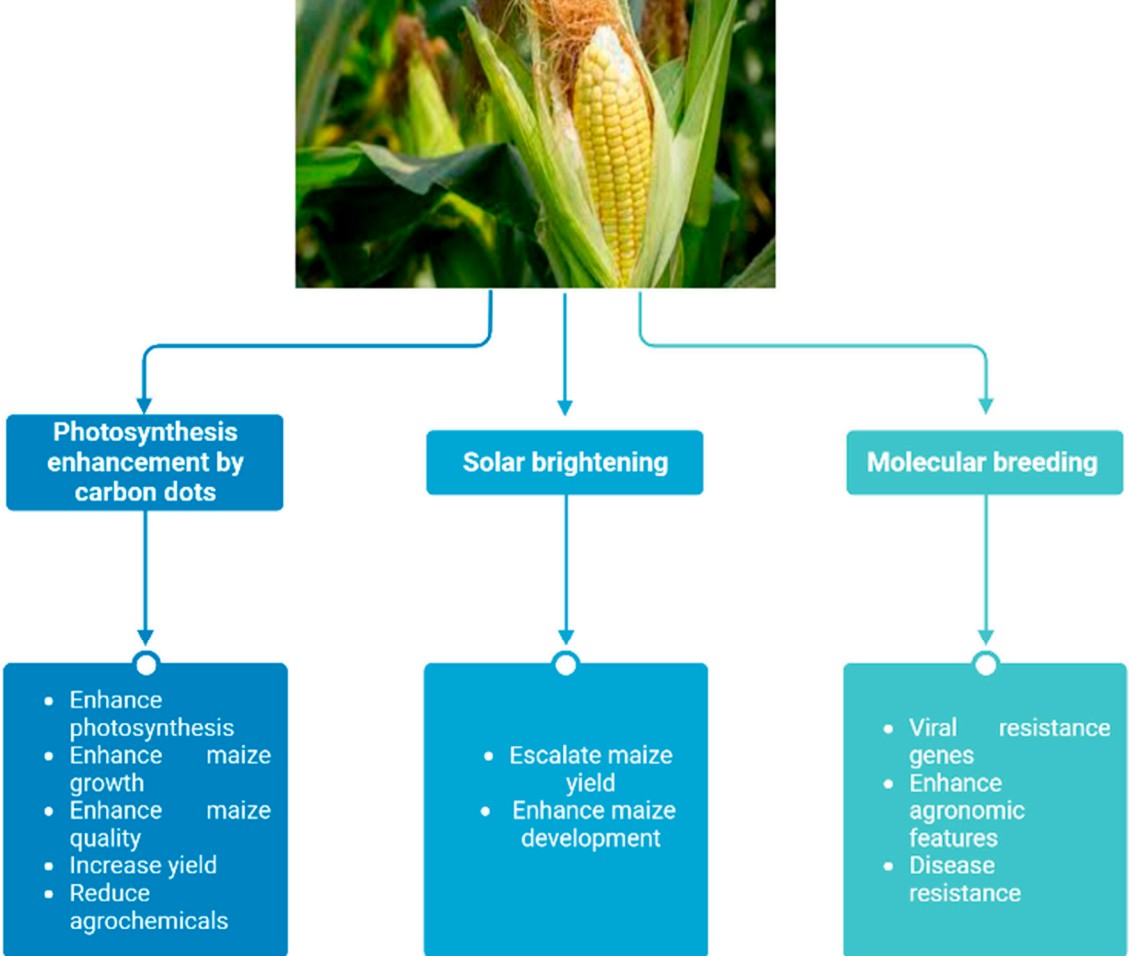

**Figure 4.** Maize-based approaches to elevate yield, growth, and crop weight.

### 5.1. Enhancing Corn Photosystem's Light Conversion Efficiency and Electron Supply Using Nitrogen-Doped Carbon Dots

It is crucial to create sustainable agricultural technologies to increase crop yields in order to feed the world's expanding population (9.7 billion by 2050). The primary metabolic activity in plants, photosynthesis, can provide all the resources needed for crop development and harvest. Therefore, enhancing photosynthesis is crucial for raising the yield of grains. Over 90% of the sunlight that reaches crops cannot be used directly, and light-harvesting nanomaterials have been proposed as promising options for increasing the light conversion efficiency in chloroplasts to enhance photosynthesis. In optoelectronic devices, chemical nano-sensors, and biological imaging, carbon dots (CDs), as metal-free fluorescent ENMs, have been used because of their great chemical stability, optical absorptivity, low toxicity, and quick photo-induced electron transfer. Additionally, CDs can be utilized as artificial antennae to increase light collection and improve photosynthesis [114,115]. Engineered nanomaterial (ENM)-based novel nano-agricultural technologies (nano-pesticides, nano-fertilizers, and nano-sensors) have demonstrated promising applications in increasing yield and quality while reducing the input of agrochemicals [116–119]. The use of CDs and nitrogen (N)-doped CDs (N-CDs), which have blue fluorescence, enhanced photosynthesis

and maize growth. Research proved that N-CDs applied topically could boost corn yield and grain weight by 15.03% and 24.50%, respectively [120].

*5.2. Improving Maize Yield by Solar Brightening*

Solar brightening (or dimming), as determined by high-quality long-term (multi-decadal) surface measurement sites, refers to the average rise (or fall) in solar energy reaching the Earth's surface for a specific area and time. The global solar brightening is estimated to be around 2 W m$^{-2}$ per decade, with regional variations ranging from as low as 0.5 W m$^{-2}$ in New Zealand to as high as 8.9 W m$^{-2}$ in Japan during the post-2000 period. With an average magnitude of about 6.6 W m$^{-2}$ per decade in the United States, surface site analysis studies also clearly demonstrated solar brightening, with one of the strongest trends in solar brightening globally [121–123]. Research showed that the rise in yield originally attributed to agricultural technology was caused by an increase in accumulated solar brightness throughout the post-flowering stage of maize development over the preceding three decades [124].

*5.3. Increasing Maize Disease Resistance through Molecular Breeding Genetic Analysis*

Disease incidence in maize is highly associated with pathogen abundance, farmed variety, weather patterns, farming practices, and agricultural ecology [125]. Pathogenic fungi are the primary cause of most maize diseases. Due to decreased yield/quality and rising input costs associated with disease control, numerous diseases result in severe economic losses. The most harmful fungi affecting maize cause foliar disease, smut, and stem/ear rot [126]. The complicated viral disease known as maize lethal necrosis (MLN), which is caused by maize chlorotic mottle virus combined with one of several viruses from the family *Potyviridae*, such as sugarcane mosaic virus (SCMV), is becoming a significant danger to the production of maize. Before maturation, maize plants are killed by the irreparable harm caused by MLN [127,128]. The output of maize is seriously threatened by maize rough dwarf disease (MRDD) on a global scale. Viruses of the genus Fijivirus in the family Reoviridae are responsible for MRDD. Only three viral disease resistance genes—ZmTrxh and ZmABP against SCMV and ZmGDI against MRDD—have been discovered and validated thus far [129,130]. The causative genes for the main QTLs Scmv1 and Scmv2 are ZmTrxh and ZmABP1, respectively, which work synergistically to provide total resistance to SCMV. The expression level of ZmTrxh, which encodes an unusual h-type thioredoxin, is highly linked with SCMV resistance. To prevent SCMV buildup without inducing salicylic acid- and/or jasmonate-mediated defensive responses, ZmTrxh is disseminated throughout the cytoplasm. It has been demonstrated that ZmABP1 cis-regulatory elements are essential for SCMV resistance because ZmABP1 encodes an auxin-binding protein and its expression level is closely correlated with disease resistance. ZmABP1 offers a second level of resistance to the rapid response mediated by ZmTrxh since it operates mostly in the later stages of viral infection [129,131,132]. Additionally, all defense pathway genes are potential targets for gene editing to boost resistance. The development of super maize varieties with high disease resistance and optimum agronomic features will only be possible using modern biotechnology, sufficient discovery of resistance genes, and knowledge of their molecular mechanisms.

## 6. Barley

A common cereal crop predominantly used for human consumption and nourishment is barley (*Hordeum vulgare* L.) [133]. *Hordeum vulgare* (barley), which has been grown in every temperate region from the Arctic Circle to the tropics, is the fourth-most important cereal crop in the world and a significant example of ecological adaptation. A member of the diploid grass family (2n = 14), barley serves as a natural model for genetics. It was among the first cereal grains to be domesticated. Wide pericentromeric regions with little to no meiotic recombination and a high proportion of repetitive elements are two characteristics of its large genome [134]. Barley has a huge 5.1 Gbp genome. Utilizing existing genetic and genomic resources, numerous accessions of both wild and domesticated barley have been

gathered and preserved to obtain various natural and artificially induced barley variations, and these accessions are essential [135]. The tremendous health benefits of barley and the strategies contributing to increased barley production are shown in Figure 5. Barley genetic management was employed to enhance barley crop yield, in which 118 barley-doubled haploid recombinants were subjected to gene-based characterization for vernalization and photoperiod. This information was used to quantify the best genotype/sowing date combination to show that avoiding extreme weather events could affect grain yield in addition to water and nitrogen management. Furthermore, because the risk of cold stress is substantially lower than the risk of heat stress, barley breeding efforts have concentrated on allelic combinations that have recessive VRN-H2 and EPS2 genes to sustain greater yields in the Mediterranean basin [136]. Recent discoveries have expanded the known function of plant MADS-box proteins in floral development by revealing the genetic factors that control barley thermo-morphogenesis and play an essential role in barley production enhancement. The barley SEPALLATA MADS-box protein, HvMADS1, is in control of preserving an unbranched spike architecture at high temperatures, whereas the loss-of-function mutant develops a branching structure resembling inflorescence. Through A-tract CArG-box motifs, HvMADS1 demonstrates enhanced binding to target promoters through temperature changes. Additionally, the cytokinin-degrading enzyme HvCKX3 is directly regulated by HvMADS1 to integrate temperature response and cytokinin homeostasis, which are necessary to inhibit meristem cell cycle/division [137]. Moreover, barley production was improved by changing the modern barley leaf architecture to have a greater vertical inclination (electrophilic canopy) so that photosynthetically active radiation could enter the crop canopy [138]. Other research showed that in order to maximize yield production in challenging environmental conditions, proper timing of plant development is essential, and the use of wild barley genes could boost grain yield formation. Using HEB-YIELD, a chosen subset of the wild barley nested association mapping population HEB-25, the effects of three abiotic stresses including nitrogen deficiency, drought, and salinity were examined. The pleiotropic effects of the Ppd-H1 gene lead to a shorter life cycle, a longer grain filling period, and larger grains in addition to the observed production increase [139]. Crops with high yields and stress tolerance are required to guarantee the availability of food in the future. When bacterial and fungal pathogen-associated molecular patterns (PAMPs) were applied, the expression of HvSTP13 increased, indicating that PAMP-triggered signaling led to transcriptional activation of the gene. In barley, the nonfunctional HvSTP13GR variation gives resistance to a biotrophic rust fungus that is commercially significant. Thus, the HvSTP13GR mutation in barley results in biotrophic resistance [140]. The top priority for all global barley breeders, after grain production, has been disease resistance. Barley is a more flexible cereal than rice and wheat and it can grow in saline, damp, dry, and higher altitude settings. However, this adaptability also broadens the variety of phytopathogens that can attack barley. Fungi, bacteria, and viruses are some of these phytopathogens [141,142]. Barley breeders have turned their focus to more advanced and integrated molecular techniques in order to create high-yielding barley varieties with improved disease resistance in order to respond quickly to these difficulties [143]. QTL mapping has been used to improve resistance to Fusarium crown rot. The effective introduction of Qcrs.cpi-1H, Qcrs.cpi-3H, and Qcrs.cpi-4H to the barley varieties Baudin, Gairdner, and Franklin increased resistance to Fusarium crown rot [144].

**Figure 5.** Genetic strategies for barley and oat yield enhancement and resistance to biotic and abiotic stresses.

### 7. Oats

Oats have a long history of use in both human and animal nutrition. The oat, which originated in the Fertile Crescent, has adapted to a variety of climate features and geographical areas [145]. Oats are healthy whole grains that contain several immuno-stimulating elements [146]. Oats (*Avena sativa* L.) suffered a significant decline in economic status during the 20th century in favor of crops with better yields, such as winter wheat and maize. Oats currently make up only 1.3% of all grains produced worldwide, and their production is dispersed. However, due to recent knowledge of its potential advantages in food, feed, and agriculture, the demand for oats is currently increasing [147]. The basic chromosomal number for wheat (Triticeae, Poaceae) and oats (*Aveneae*) is x = 7. The oat genome frequently exhibits intergenomic translocations, in contrast to the wheat genome, which exhibits similar translocations less frequently [148]. The domesticated oat, *Avena sativa* L., is an allohexaploid (AACCDD, 2n = 6x = 42), and it is believed to have first appeared in Anatolia's wheat, emmer, and barley fields more than 3000 years ago as a weed. Oats are one of the food crops and ancient grains that are grown and consumed all over the world. It is becoming increasingly popular due to its nutritional makeup and the numerous advantages of certain bioactive ingredients [14]. Oats have the potential to replace foods including animal products due to their minimal carbon footprint and significant health advantages [149]. Oats (*Avena sativa* L.) are rich in antioxidants, minerals, vitamins, and dietary fiber (such as β-glucans). Recent findings investigated the production, grain quality, and metabolite responses of four oat types to elevated nitrogen levels over several sites and years. The levels of amino acids, total protein, and lipids containing nitrogen were all boosted by nitrogen supplementation, according to the combined phenotyping approach. Thus, adding nitrogen to grains significantly boosted grain production and β-glucan content [150]. Due to increased water scarcity, the sustainability and profitability of oats crops in a Mediterranean agroecosystem were studied in the SUPROMED project (sustainable production in water-limited environments of Mediterranean agroecosystems). To assess the effects of MOPECO (model for the economic optimization of irrigation water use at the farm level), the irrigation scheduling model built within the SUPROMED platform available to farmers, several productive, financial, and environmental key performance indicators (KPIs) were examined. When compared to conventional management, the SUPROMED management method increased oat yield while using 40% less water [151]. However, oats with Fusarium disease have lower quality yields because of decreased germination due to contamination of the grain with mycotoxins. In comparison to the control variety Vyatskiy, two oat breeding lines 54h2476 and 66h2618 as well as novel

variety Azil (57h2396) demonstrated high resistance to Fusarium fungus infection and mycotoxin contamination [152].

## 8. Grain Varietal Improvement

Most small-scale farmers employ conventional crop varieties in most developing nations, which have low yields and may be more susceptible to drought, heat, diseases, and other pressures. Better quality, more consistent production, and substantially higher yields are all features of modern enhanced cultivars. More than 880 new types have been made available for farming, and they produce annual benefits worth USD 850 million. To maximize yields from the new varieties, technology 'packages' have been created for crop management, irrigation, and pest control [153]. To create perennial substitutes with the necessary traits—larger seed size, stronger stems, enhanced palatability, and higher seed yield—varietal development is required. It may be possible to increase soil structure, carbon sequestration, water, nutrient retention, and yields by breeding plants with deeper and bushier root systems. Increased root mass is thought to have the potential to store between 5 and 10 $kg/m^2$ (50–100 tons/ha) of carbon, which would have a globally significant impact on atmospheric carbon sequestration. With the use of new biotechnologies, particularly those involving genome editing (such as Clustered Regularly Interspaced Palindromic Repeats, or CRISPR), crop varietal improvement will become more efficient and more narrowly focused [154]. From the Kalyan sona and RR21 types, three wheat varieties, Kudrat 5, Kudrat 9, and Kudrat 17, were created. The plant heights of Kudrat 9, Kudrat 5, and Kudrat 17 were respectively 85–90, 95–100, and 90–95 cm, while the spike lengths were 9, 6, and 10 cm. The weights of 1000 seeds were 70–72, 58–60, and 60–62 g, respectively, and the yields per acre were 20–25, 15–20, and 22–27 quintals [155]. According to the biplot analysis, the best rice varieties at both locations in the minor season were WITA 9, GT 11, ARS-957-BGJ-171-15-D-B, NERICA L36, and AGRA, whereas the best varieties in the major season were FARO 66, SAHEL 317, and Amankwatia, for which the production potential was around 92% higher than the output of the local checks [156]. Additionally, elite heat-tolerant maize varieties are being developed and planted in South Asia, which is a significant step forward. Protecting smallholder maize crops from the changing climates in the Sub-Sahara and Asia requires more genetic gain in grain production in stress-prone settings, together with quicker replacement of old/obsolete varieties, through intense engagement with seed companies [157].

## 9. Potential Challenges and Future Perspectives

Plants have a complex, multi-layered immune system to protect themselves against pests and pathogens because of long-term co-evolution with their pathogens. Despite this, there are a few significant issues that must be resolved to successfully mitigate various biotic stressors. Rust-causing pathogens continue to rapidly evolve into new strains. It is widely known that the rust pathogens exhibit high pathogenic diversity and attempts to develop long-lasting resistance to these diseases have been hampered by the regular appearance of new, virulent strains that outcompete resistance genes found in cultivated varieties [158]. To minimize losses from attacks by rapidly evolving and more virulent pathogenic races, it is crucial to breed wheat for durable and broad-spectrum disease resistance at a faster rate in a changing global climate. Additionally, this would result in less pesticide (fungicide) use, thereby avoiding risks to the environment and human health, which is a crucial element of contemporary sustainable crop production systems.

## 10. Conclusions

Considerable efforts are required to fulfill the expanding food demands due to the expected rise in the global population by 2050. To address the global dilemma of food availability, we must change our mindset and way of life, invest more in knowledge development, and promote the use of cutting-edge technology that can accelerate the process of achieving these objectives. Long-term success, free access to innovative technologies, and

research are critical for preserving various creative yield-enhancing techniques with documented advantages for genetic yield gain and yield stability in vital daily-needed crops. Thus, the indispensable crop yield-enhancing methods described in this paper include strategies such as bacterial-nutrient management, solar brightening, fertilizer management, reducing harmful gas emissions, enhancing photosynthesis, stress tolerance, disease resistance, and varietal improvement to increase grain production and plant resistance to harmful environmental factors. However, red light management and genetic manipulations propose the most grain yields by approximately double. Thus, these approaches can promisingly boost the adaptability of sustainable agroecosystems by elevating production levels, lessening the negative effects of climate change, and supporting food security as well as everyone's health.

**Author Contributions:** Conceptualization, G.A. and A.H.; investigation, G.A. and M.M.; writing-review and editing, G.A. and E.B.; visualization, E.B., M.M. and A.B.; project administration, A.A.A., A.H. and M.N. All authors have read and agreed to the published version of the manuscript.

**Funding:** This research received no external funding.

**Data Availability Statement:** This paper has all the data supporting the findings.

**Acknowledgments:** G.A., A.H, and M.N. Lebanese University faculty of sciences.

**Conflicts of Interest:** The authors declare no conflict of interest.

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
