# Peer review of "Enhancing Essential Grains Yield for Sustainable Food Security and Bio-Safe Agriculture through Latest Innovative Approaches"

_agronomy, doi:10.3390/agronomy13071709_

Round 1

Reviewer 1 Report

This review is valuable and presents the actual stage of molecular technology and genetics for improving the production of the leading grain crops (wheat, rice, maize, barley, and oat). 

I have some observations.

Introduction Need citation for "Maize kernels can be white, yellow, orange, red, or black depending on environmental, cultural, and genetic factors." (rows 64-65)

Need citation for "Barley can be further divided into variants with normal, waxy, or high amylose starch, high lysine, high beta-glucan, and proanthocyanin-free grains based on the grain’s composition." (rows 70-71)

Need citation for "In addition to being a significant source of dietary fiber and a rich source of pro- tein, oats also contain several vital minerals, lipids, and the mixed-linkage polysaccharide- glucan." (rows 75-76).

2. Wheat

Need citation for "The most widely cultivated crop on the planet, wheat (Triticum aestivum L.), contributes nearly a fifth of the total calories consumed by people and has the highest protein content of any meal." (rows 94-95).

Need citation for "In this continuous period of climate change, in addition to the rapidly expanding population, biotic stresses and escalating abiotic pressures pose a significant threat to wheat production all over the world." (rows 110-112)

Why are not information concerning progress in wheat disease resistance ?

3. Rice

Need more citations for "More than one third of the world’s population, 90% of whom live below the poverty line, depend on rice as a major food source." (rows 229-231).

Rows 234, replace "illness" with "diseases"

Need more citations for "A novel strategy for sustainable agriculture may involve the use of substances that control plant development and have the potential to boost crop yields and reduce CH4 emissions." (rows 270-272)

Need citations for "The development of ethephon (2-Chloroethylphosphonic acid), a precursor of ethyleneC2H4, as a potential soil amendment for lowering methane (CH4) emissions was studied." (rows 272-274).

Need citations for "Although it depends on the type of organic system, organic agriculture has been suggested to lessen harmful effects on the environment and human health." (rows 289-290).

Need citations for "It has been demonstrated that the integration of Azolla, fish, and ducks produces durable systems that retain high yields under heavy rainfall. Additionally, research demonstrated that the rice cropping system’s ability to adapt to harsh weather events was improved by adding additional features." (rows 295-298)

Again no information concerning resistance to rice diseases or pests. 

The same to Corn, Barley, or Oats. No information concerning the molecular approach to resistance to plant diseases or pests. 

English language is ok

Only minor modifications are necessary 

Author Response

Dear Reviewers, Dear Editor,

Thank you so much for your time, consideration, and on-point remarks, all taken into consideration.

The whole manuscript was revised.

Hope the revised version of the manuscript matches all the criteria to be published in your honorable journal as a review paper.

Reviewer 1 :

Q : This review is valuable and presents the actual stage of molecular technology and genetics for improving the production of the leading grain crops (wheat, rice, maize, barley, and oat). 

A : Dear reviewer,

Thank you so much for your comments.

All are revised.

Q : I have some observations.

Introduction Need citation for "Maize kernels can be white, yellow, orange, red, or black depending on environmental, cultural, and genetic factors." (rows 64-65)

A : Thank you for your comment, the citation is added.

Q : Need citation for "Barley can be further divided into variants with normal, waxy, or high amylose starch, high lysine, high beta-glucan, and proanthocyanin-free grains based on the grain’s composition." (rows 70-71)

A : Thank you for your comment, the citation is added.

Q : Need citation for "In addition to being a significant source of dietary fiber and a rich source of pro- tein, oats also contain several vital minerals, lipids, and the mixed-linkage polysaccharide- glucan." (rows 75-76).

A : Thank you for your comment, the citation is added.

Q : 2. Wheat

Need citation for "The most widely cultivated crop on the planet, wheat (Triticum aestivum L.), contributes nearly a fifth of the total calories consumed by people and has the highest protein content of any meal." (rows 94-95).

A : Thank you for your comment, the citation is added.

Q : Need citation for "In this continuous period of climate change, in addition to the rapidly expanding population, biotic stresses and escalating abiotic pressures pose a significant threat to wheat production all over the world." (rows 110-112)

A : Thank you for your comment, the citation is added.

Q : Why are not information concerning progress in wheat disease resistance ?

A : Thank you for your comment, more information are added for wheat disease resistance.

Utilizing genetic rust resistance in wheat and integrated rust management techniques to cre                             ate more resilient cultivars.

Fungi and insects reduce global wheat production by 21.5% each year[52]. Rust diseases, which are among the most economically significant diseases impacting wheat production, are caused by biotrophic pathogenic fungus. The rust pathogens Puccinia graminis f. sp. tritici, Puccinia triticina (pt), and Puccinia striiformis f. sp. tritici , which cause stem rust, leaf rust , and stripe rust , respectively,  put the world's wheat production at risk all year[53–55]. Wheat leaves at different developmental stages, leaf sheath, and glumes are the main targets of pt infection. Several different genes known as Lr, Yr, and Sr, respectively, give resistance to leaf rust, stripe rust, and stem rust[56,57]. However, despite significant advancements in the treatment of wheat diseases created by scientific and technological innovations, plant diseases continue to pose serious dangers to the world's wheat output.[58,59]. More thorough understanding of the spatial and temporal heterogeneity in the growth pattern of wheat rust must contribute to more efficient and long-lasting disease control[60]. However, the ability to anticipate the severity of wheat leaf rust from its interactions with the climate (relative humidity and temperature), the disease (onset and severity), the genotype (maturity and resistance), and the planting date is still limited[61]. For the establishment of quantitative and mature plant resistance, wheat TaWAK6, a non-arginine-aspartate wall-associated kinase with an extracellular GUB domain, a calcium-binding epidermal growth factor domain, and a cytoplasmic serine/threonine kinase domain, was demonstrated to be significant[62]. TaRPM1, an NBS-LRR gene in wheat, has demonstrated that it favors the salicylic acid signaling system, which confers Pst resistance in high-temperature seedling plants.  During plant-pathogen interactions, silicon-induced biochemical or molecular resistance predominated, involving the activation of defense-related enzymes, regulation of the intricate network of signal pathways, stimulation of the production of antimicrobial compounds, and activation of defense-related genes[63,64]. Wheat stripe rust treatment techniques have advanced significantly, but more research is needed to understand how this pathosystem's environment, disease, and planting date interactions affect resistance[65]. Because planting date has also been regarded as a key disease management strategy in a wide range of agricultural crops, the close association between wheat maturation and wheat leaf rust predictions seems significant. Furthermore, given that a combination of environment, maturity, and planting date would be investigated for screening wheat varieties, these findings may improve breeding for more resistant genotypes[66].These connections need to be further investigated in other geographic regions with different host and pathogen genotypes, climatic conditions, and environmental situations.

 Q : 3. Rice

Need more citations for "More than one third of the world’s population, 90% of whom live below the poverty line, depend on rice as a major food source." (rows 229-231).

A : Thank you for your comment, more citations are added.

Q : Rows 234, replace "illness" with "diseases"

A : Thank you, it is replaced.

Q : Need more citations for "A novel strategy for sustainable agriculture may involve the use of substances that control plant development and have the potential to boost crop yields and reduce CH4 emissions." (rows 270-272)

A : Thank you for your comment, more citations are added.

Q : Need citations for "The development of ethephon (2-Chloroethylphosphonic acid), a precursor of ethyleneC2H4, as a potential soil amendment for lowering methane (CH4) emissions was studied." (rows 272-274).

A : Thank you for your comment, more citations are added.

Q : Need citations for "Although it depends on the type of organic system, organic agriculture has been suggested to lessen harmful effects on the environment and human health." (rows 289-290).

A : Thank you for your comment, the citation is added.

Q : Need citations for "It has been demonstrated that the integration of Azolla, fish, and ducks produces durable systems that retain high yields under heavy rainfall. Additionally, research demonstrated that the rice cropping system’s ability to adapt to harsh weather events was improved by adding additional features." (rows 295-298)

A: Thank you for your comment, more citations are added.

Q : Again no information concerning resistance to rice diseases or pests. 

The same to Corn, Barley, or Oats. No information concerning the molecular approach to resistance to plant diseases or pests. 

A : Thank you for you comments, molecular approaches for rice, corn, barley and oats resistance to diseases have been added

3.6. Increasing rice blast disease resistance by molecular breeding approach

The most significant constraints on rice production are diseases. There have been reports of over 70 diseases affecting rice that are brought on by fungi, bacteria, viruses, or nematodes [103]. The most severe rice disease is rice blast (Magnaporthe oryzae), which is particularly destructive when conductive conditions are present and since there have been numerous outbreaks of the blast disease in rice, attempts have been made to create new cultivars that are resistant to the disease[103,104]. To create an integrated management program for blast resistance, strategic research is currently focused on bridging the gaps in our understanding of biotic stresses on rice, particularly by advancing our understanding of the molecular genetics of blast disease.  Fine mapping's actual identification of potential DNA markers associated to resistance genes may enable rice breeders to effectively transfer these genes from donor cultivars into new, elite rice cultivars via marker-assisted selection (MAS)[105]. There are many uses for allele mining in grain improvement, including allele identification, allelic variation characterization, haplotype identification, analysis of haplotype diversity between related or different haplotypes of the same gene, evolutionary relationship, and similarity analysis, as well as the creation of molecular markers to distinguish one allele from other alleles[106]. The goal of allele mining is to find the best alleles for blast resistance in rice species that are both wild and domesticated. Numerous important blast resistance genes have so far been found to have unique and superior alleles in both wild and farmed rice varieties using mining techniques. The resistance and susceptible alleles of Pi54 have been distinguished using functional markers generated from allele mining approaches, which also examined the sequence level similarity for the Pikm alleles. Allele mining has also been used to differentiate M. oryza from M. grisea[107,108].

4.3. Increasing Maize disease resistance by molecular breeding genetic analysis

Disease incidence in maize is highly associated with pathogen abundance, farmed varieties, weather patterns, farming practices, and agricultural ecology[124]. Pathogenic fungi are the primary cause of most maize diseases. Due to decreased yield/quality and rising input costs associated with disease control, numerous diseases result in severe economic losses. Among the most harmful fungi that affect maize are foliar disease, smut, and stem/ear rot[125]. The complicated viral disease known as maize lethal necrosis (MLN) which is caused by maize chlorotic mottle virus combined with one of several viruses from the Potyviridae, such as sugarcane mosaic virus (SCMV), is becoming a significant danger to the production of maize. Before they mature, maize plants are killed by the irreparable harm caused by MLN[126,127]. The output of maize is seriously threatened by maize rough dwarf disease (MRDD) on a global scale.Viruses in the Fijivirus genus of the Reoviridae family are responsible for MRDD. Only three viral disease resistance genes—ZmTrxh and ZmABP against SCMV and ZmGDI against MRDD—have been discovered and validated thus far [128,129]. The causative genes for the main QTLs Scmv1 and Scmv2, respectively, are ZmTrxh and ZmABP1, which work synergistically to provide total resistance to SCMV. The expression level of ZmTrxh, which encodes an unusual h-type thioredoxin, is highly linked with SCMV resistance.To prevent SCMV buildup without inducing salicylic acid- and/or jasmonate-mediated defensive responses, ZmTrxh is disseminated throughout the cytoplasm. It has been demonstrated that ZmABP1 cis-regulatory elements are essential for SCMV resistance because ZmABP1 encodes an auxin-binding protein and that its expression level is closely correlated with disease resistance. ZmABP1 offers a second level of resistance to the rapid response mediated by ZmTrxh since it operates mostly in the later stages of viral infection[128,130,131]. Additionally, all defense pathway genes are potential targets for gene editing to boost resistance. The development of super maize varieties with high disease resistance and optimum agronomic features will only be possible through the use of modern biotechnology, sufficient discovery of resistance genes, and a knowledge of their molecular mechanisms.

Molecular Approach for barley disease resistance :

The top priority for all global barley breeders, after grain production, has been disease resistance. Barley is a more flexible cereal than rice and wheat and can grow in saline, damp, dry, and higher altitude settings. But this adaptability also broadens the variety of phytopathogens that attack barley. Fungi, bacteria, and viruses are some of these phytopathogens[143,144]. Barley breeders turned their focus to more advanced and integrated molecular techniques in order to create high-yielding barley varieties with improved disease resistance in order to respond quickly to the difficulties[145]. QTL mapping is used to improve resistance to Fusarium crown rot. The effective introduction of Qcrs.cpi-1H, Qcrs.cpi-3H, and Qcrs.cpi-4H to the barley varieties Baudin, Gairdner, and Franklin  increased resistance to Fusarium crown rot[146].

Molecular approach for oats disease resistance :

However, oats with Fusarium disease have lower quality yields because of decreased germination led on by Fusarium fungi's contamination of grain with mycotoxins .In comparison to the control variety Vyatskiy, two oat breeding lines, 54h2476 and 66h2618, as well as a novel variety, Azil (57h2396), can be described for having a high resistance to Fusarium fungus infection and mycotoxin contamination[154].

Regards

Reviewer 2 Report

RE: Enhancing Essential Crops Yield for Sustainable Food Security and Bio-Safe Agriculture through Latest Innovative Approaches

The general structure of the manuscript is not clear. he lack of a strong conceptual framework weakens the overall impact of the manuscript. It is well designed for a book chapter. Keeping in view the  rigor of writing, I am unable to accept this for publication as review paper in the journal. Unless providing a proper review, which take into account the quantitative description of the different approaches, this manuscript do not full fill the requirements for this journal.

However, with substantial revisions addressing the issues raised as major and minor comments  in addition to considering all the points like the review of literature has been taken from where, what were reviewed, how was reviewed, what aspect was considered, what was the selection criteria of the articles in addition, the revised manuscript has the potential to make a meaningful contribution to the field can re-evaluated for possible publication as review paper.

Major comments

1.       In abstract, the most promising approaches, which has been given in the body of the ms, should be written and its comparative effectiveness should be documented here in the abstract, this is a key issue in the abstract where no results have been provided.

2.       The introduction section needs drastic revision, only one paragraph introduction is provided in the articles, without highlighting the missing gap and possible solution for the missing gap, also have no objective though it is review, but still we need to learn, why this review is going to be conducted.

3.       The introduction section should be split several parts like first describing the problem/scope of low productivity, followed by available production and missing gap, how much we need, then how these can be obtained while describing some of the new and recent approaches, and then justification/objective of documenting this review.

4.       Fig 1, this is not clear, we can get no information form here, how much was the impact of climate etc, what were the different sustainable approaches------------ we are not getting any information from here, this should be mixed with literature quantitatively

5.       I do not see any methodological approach, that how the review of literature was taken, which aspect was studied/reviewed and what were their reports and results

6.       The methodology employed in this study is inadequately described or totaly missing, making it challenging to assess the validity and reliability of the findings presented. Important details such as the selection criteria for the innovative approaches, data collection methods, and statistical analysis techniques are either missing or insufficiently elaborated upon. Without these crucial methodological details, the scientific rigor of the study is questionable

7.       The information about each crop should be squeezed, in term of evolution, but more stress should be given on the annotative approach description, and its effects on crop production quantitatively

8.       Fig 2 (all other figures, which approaches has been mentioned)  the order of the approaches should be in proper sequences, with quantitative description of increase and proper reference to each approach , what about the hybrid wheat production --------- this is not a new approach

9.       Fig 3, all the approaches which has been explained should be provided here in this figure, along with some quantitative description and reference

10.   Figure 4. No innovative techniques has been provided for maize production, please revisit the data base and include some of the newest techniques for maize production.

11.   For barley and oat, you need to present the recent approaches in pictorial illustration

12.   The manuscript lacks critical evaluation and discussion of the limitations and potential challenges associated with the innovative approaches discussed. While some limitations are briefly mentioned, they are not adequately explored, and no effort is made to address them or propose possible solutions. This omission weakens the practical applicability and relevance of the study.

13.   The conclusion, section does not show what were the methods for crops, and which one was better based on the available literature been reviewed in this manuscript. Here we need a clear take home message. It does not offer a comprehensive analysis of the latest innovative approaches.

14.    

Minor comments

What do you mean by essential crop yield?--------- replace with a more suitable term like grains or cereals etc.

L16, A key concern in agriculture is how to feed the expanding population while reducing the effects of climate change---------------- do you mean by this, you mean effect of climatic changes is the risk of food security, this statement should be like “A key concern in agriculture is how to feed the expanding population and safeguard the environment from the ill effects of climate change” my suggestion

L20, rising demand ----------- rephrase like rising demand of food.

L21, However--------- replace with thus

L37, The COVID-19 pandemic --------------------natural disasters, war, and violence------------ please provide a valid reference

L79, innovative yield-increasing ------------- agricultural strategies (Figure 1)---------------- no approaches is given in the Fig 1

L110, In this continuous period of climate change, in addition to -----------------to wheat production all over the world----------------- these aspect should be highlighted quantitively with the help of literature, --------------these just vague statement and not fit for review)

L124, environmentally benign------------ please try to use more acceptable terminology like environmentally friendly etc

L511, Agriculture has always been concerned with crop yield and yield stability. Nonethe- 511 less, c-------------- please remove no need of this statement

L514, illnesses are destroying nearly half of the world’s grain crop--------------- this aspect has not been reviewed

L514, biodiversity is dwindling at an alarming rate------------------ not reviewed

L515, Our planet is 515 confronting unprecedented problems--------------- which problems ------ not reviewed in the ms

L519, Contrarily, agriculture and food production must improve their sustainability.------------ delete 

Nil 

Author Response

Dear Reviewers, Dear Editor,

Thank you so much for your time, consideration, and on-point remarks, all taken into consideration.

The whole manuscript was revised.

Hope the revised version of the manuscript matches all the criteria to be published in your honorable journal as a review paper.

Reviewer 2 :

Q: RE: Enhancing Essential Crops Yield for Sustainable Food Security and Bio-Safe Agriculture through Latest Innovative Approaches

The general structure of the manuscript is not clear. he lack of a strong conceptual framework weakens the overall impact of the manuscript. It is well designed for a book chapter. Keeping in view the  rigor of writing, I am unable to accept this for publication as review paper in the journal. Unless providing a proper review, which take into account the quantitative description of the different approaches, this manuscript do not full fill the requirements for this journal.

However, with substantial revisions addressing the issues raised as major and minor comments  in addition to considering all the points like the review of literature has been taken from where, what were reviewed, how was reviewed, what aspect was considered, what was the selection criteria of the articles in addition, the revised manuscript has the potential to make a meaningful contribution to the field can re-evaluated for possible publication as review paper.

A: Dear reviewer,

Thank you so much for your comments.

All major and minor comments are addressed in the revised manuscript.

Major comments

Q1.       In abstract, the most promising approaches, which has been given in the body of the ms, should be written and its comparative effectiveness should be documented here in the abstract, this is a key issue in the abstract where no results have been provided.

A : Thank you for your comment, the abstract is revised so that addressed approaches are documented in the abstract.

Q2.       The introduction section needs drastic revision, only one paragraph introduction is provided in the articles, without highlighting the missing gap and possible solution for the missing gap, also have no objective though it is review, but still we need to learn, why this review is going to be conducted.

A: Thank you for your comment,

The introduction section is revised.

Q3.       The introduction section should be split several parts like first describing the problem/scope of low productivity, followed by available production and missing gap, how much we need, then how these can be obtained while describing some of the new and recent approaches, and then justification/objective of documenting this review.

A: Thank you for your comment.

The introduction is all revised and split into several paragraphs, documenting the problem, the gap, how the approaches are incorporated in solving the problem, and justifying the review objective.

Q4.       Fig 1, this is not clear, we can get no information form here, how much was the impact of climate etc, what were the different sustainable approaches------------ we are not getting any information from here, this should be mixed with literature quantitatively

A: Thank you for the comment, more information is added to figure 1.

Q5.       I do not see any methodological approach, that how the review of literature was taken, which aspect was studied/reviewed and what were their reports and results

A: Thank you for your comment,

The methodological approach was added.

Q6.       The methodology employed in this study is inadequately described or totaly missing, making it challenging to assess the validity and reliability of the findings presented. Important details such as the selection criteria for the innovative approaches, data collection methods, and statistical analysis techniques are either missing or insufficiently elaborated upon. Without these crucial methodological details, the scientific rigor of the study is questionable

A: Thank you so much for your comment.

The methodological section is added.

Q7.       The information about each crop should be squeezed, in term of evolution, but more stress should be given on the annotative approach description, and its effects on crop production quantitatively

A: Thank you for your comment,

Information on each crop has been lessened, and more data is added concerning the innovative approach.

Q8.       Fig 2 (all other figures, which approaches has been mentioned)  the order of the approaches should be in proper sequences, with quantitative description of increase and proper reference to each approach , what about the hybrid wheat production --------- this is not a new approach

A: Thank you for the comment, approaches are properly sequenced with quantitative description.

Q9.       Fig 3, all the approaches which has been explained should be provided here in this figure, along with some quantitative description and reference

A: Thank you for the comment, the other explained approaches are added.

Q10.   Figure 4. No innovative techniques has been provided for maize production, please revisit the data base and include some of the newest techniques for maize production.

A: Thank you for the comment, innovative techniques for maize production have been added to figure 4.

Q11.   For barley and oat, you need to present the recent approaches in pictorial illustration

A: Thank you for your comment, the recent approaches of barley and oats are presented in same pictorial illustration.

Q12.   The manuscript lacks critical evaluation and discussion of the limitations and potential challenges associated with the innovative approaches discussed. While some limitations are briefly mentioned, they are not adequately explored, and no effort is made to address them or propose possible solutions. This omission weakens the practical applicability and relevance of the study.

A: Thank you for your comment,

Potential challenges and future perspectives or solutions are added.

Q13.   The conclusion, section does not show what were the methods for crops, and which one was better based on the available literature been reviewed in this manuscript. Here we need a clear take home message. It does not offer a comprehensive analysis of the latest innovative approaches.

A: Thank you for your comment,

The conclusion is revised.

Minor comments

Q: What do you mean by essential crop yield?--------- replace with a more suitable term like grains or cereals etc.

A: Thank you for the comment, crops is replaced by grains as much as possible throughout the whole manuscript.

Q: L16, A key concern in agriculture is how to feed the expanding population while reducing the effects of climate change---------------- do you mean by this, you mean effect of climatic changes is the risk of food security, this statement should be like “A key concern in agriculture is how to feed the expanding population and safeguard the environment from the ill effects of climate change” my suggestion

A: Thank you for the comment, it is replaced.

Q:L20, rising demand ----------- rephrase like rising demand of food.

A: Thank you, it is added.

Q: L21, However--------- replace with thus

A: Thank you it is removed.

Q:L37, The COVID-19 pandemic --------------------natural disasters, war, and violence------------ please provide a valid reference

A: Thank you, it is removed.

Q: L79, innovative yield-increasing ------------- agricultural strategies (Figure 1)---------------- no approaches is given in the Fig 1

A: Thank you for the comment, approaches are demonstrated in the review objective.

Q: L110, In this continuous period of climate change, in addition to -----------------to wheat production all over the world----------------- these aspect should be highlighted quantitively with the help of literature, --------------these just vague statement and not fit for review)

A: Thank you for the comment, this vague statement is removed.

Q: L124, environmentally benign------------ please try to use more acceptable terminology like environmentally friendly etc

A: Thank you, it is replaced.

Q: L511, Agriculture has always been concerned with crop yield and yield stability. Nonethe- 511 less, c-------------- please remove no need of this statement

A: Thank you for the comment, it is removed.

Q: L514, illnesses are destroying nearly half of the world’s grain crop--------------- this aspect has not been reviewed

A: Thank you for the comment, it is removed.

Q: L514, biodiversity is dwindling at an alarming rate------------------ not reviewed

A: Thank you , it is removed.

Q: L515, Our planet is 515 confronting unprecedented problems--------------- which problems ------ not reviewed in the ms

A: Thank you, it is removed.

Q: L519, Contrarily, agriculture and food production must improve their sustainability.------------ delete 

A: Thank you, it is deleted.

Regards

Reviewer 3 Report

This review focused on how to feed the expanding population under climate change. And the key most up-to-date approaches that boost the world's most widely cultivated food crops such as corn, wheat, and rice had been summarized. The knowledge presented in this review revealed that the usage of the methods described in this paper including fertilizers management and formulations, smart agricultural activities, complex agricultural systems, and biotechnological genetic approaches can promisingly boost the adaptability of sustainable agroecosystems, lessen the negative effects of climate change. However, this review is lack the content of the varieties improvement which is very important for crop yield promotion.

Author Response

Dear Reviewers, Dear Editor,

Thank you so much for your time, consideration, and on-point remarks, all taken into consideration.

The whole manuscript was revised.

Hope the revised version of the manuscript matches all the criteria to be published in your honorable journal as a review paper.

Reviewer 3 :

Q : This review focused on how to feed the expanding population under climate change. And the key most up-to-date approaches that boost the world's most widely cultivated food crops such as corn, wheat, and rice had been summarized. The knowledge presented in this review revealed that the usage of the methods described in this paper including fertilizers management and formulations, smart agricultural activities, complex agricultural systems, and biotechnological genetic approaches can promisingly boost the adaptability of sustainable agroecosystems, lessen the negative effects of climate change. However, this review is lack the content of the varieties improvement which is very important for crop yield promotion.

A: Dear reviewer,

Thank you so much for your comments.

More content has been added concerning varieties improvement in the revised manuscript.

Regards

Round 2

Reviewer 1 Report

After I read the manuscript, after the corrections of the author,  I am agree with the publishing of the manuscript in present form. 

English language is ok. 

Reviewer 2 Report

The Authors have incorporated most of my comments, and thus I accept the manuscript in present condition. 

The English writing is fine and acceptable for publication